# An electrochemical biosensor for the detection of *Mycobacterium tuberculosis* DNA from sputum and urine samples

**Daniel Ramos-Sono[1], Raúl Laureano[1], Daniel Rueda[1], Robert H. Gilman[2], Adolfo La Rosa[3], Jesús Ruiz[4], Raúl León[4], Patricia Sheen[1], Mirko Zimic[1]***

**1** Laboratorio de Bioinformática y Biología Molecular, Laboratorios de Investigación y Desarrollo, Facultad de Ciencias y Filosofía, Universidad Peruana Cayetano Heredia, Lima, Perú, **2** Department of International Health, Johns Hopkins Bloomberg School of Public Health, Baltimore, Maryland, United States of America, **3** Laboratorio de Electroquímica, Facultad de Ciencias, Universidad Nacional de Ingeniería, Lima, Perú, **4** Laboratorio de Metalurgia y Ciencias de Materiales, NDT Innovations, Inc., Lima, Perú

* mirko.zimic@upch.pe

**Data Availability Statement:** All relevant data are within the manuscript and its Supporting Information files.

## Abstract

Tuberculosis (TB) is a major global public health problem with high mortality and morbidity. In low-middle income countries (LMIC) a large number of respiratory symptomatic cases that require TB screening per year demands more accurate, fast and affordable testing for TB diagnostics. Sputum smear is the initial screening test in LMICs, however, its sensitivity is limited in patients with low sputum bacilli load. The same limitation is observed in the currently available molecular tests. We designed, standardized and evaluated an electrochemical biosensor that detects the highly specific DNA insertion element 6110 (IS6110). A PCR amplified DNA product is hybridized on the surface of the working electrode built on FTO-Glass with immobilized specific DNA probes, after which cyclic voltammetry is performed with an Ag/AgCl reference electrode and a platinum counter electrode. The response of the sensor was measured by the ratio (cathodic peak current of the hybridized sensor) / (cathodic peak current of the non-hybridized sensor). We tested the biosensor, using positive hybridization control sequences, genomic DNA extracted from *M. tuberculosis* strains and sputum of TB patients, and extracted DNA from the urine of healthy controls spiked with *M. tuberculosis* DNA. This biosensor was effective for the detection of *M. tuberculosis* DNA with a detection limit of 16 fM in sputum sample and 1 fM in spiked urine samples. The low cost and the relatively brief duration of the assay make this an important TB screening tool in the fight against tuberculosis.

## Introduction

Tuberculosis (TB) continues to represent a problem in global public health. The World Health Organization (WHO) released a 2015 report that estimated around 9.6 million new cases of TB; including 1 million children. It also reported 1.5 million deaths attributed to this illness, with 400,000 deaths in HIV positive patients, revealing that the mortality associated with TB is

**Funding:** This study was funded by the Wellcome Trust (Ref: 99805/Z/12/Z (https://wellcome.ac.uk/grant-funding) awarded to PS, and partly by the Grand Challenge Canada GCC Number 0687-01-10 (https://www.grandchallenges.ca) awarded to PS. This study was also supported by the ERANET LAC ELAC2014/HID-0352 (https://www.era-learn.eu/network-information/networks) awarded to MZ, NIH Grant R25 TW009720-01 awarded to MZ, and Google LATAM award 2016 awarded to MZ. NDT Innovations Inc, provided the salary of JR and RL. he specific roles of these authors are articulated in the 'author contributions' section. NDT Innovations also provided access to the electron microscopy equipment but did not have any additional role in the study design, data collection and analysis, decision to publish, or preparation of the manuscript.

**Competing interests:** The authors of this paper have read the journal's policy and have the following competing interests: JR and RL are paid employees of NDT Innovations Inc. There are no patents, products in development or marketed products associated with this research to declare. This does not alter our adherence to PLOS ONE policies on sharing data and materials.

unacceptably high considering the available diagnostics and effective treatment that can cure the majority of patients [1].

TB infection is acquired from aspiration of Mycobacterium tuberculosis, this pathogen is an obligate aerobic bacteria [2], that produces a bacterial proliferative phase, and a subsequent immunologic response against the bacteria pushing TB into a latent phase without apparent symptoms. Patients in the latent stage or the primary stages of the illness are hard to diagnose and represent a reservoir of TB in the community [3]. Currently, there are multiple methods for detecting TB, with cost, sensitivity, specificity, duration of the test, and operator training being the primary limitations of these techniques.

The classic methods such as Ziehl-Neelsen staining and the culture of samples obtained from patients are relatively easy to carry out, but they have important disadvantages. Staining is highly specific but has a variable modest sensitivity. While culture growth usually takes a long time from 10 to 60 days [4–6]. DNA-based molecular techniques, using polymerase chain reaction (PCR) are some of the most promising, especially in the detection of multi-drug resistant TB [7, 8]. DNA fragments are amplified with PCR and analyzed by electrophoresis to verify amplification. These tests require a specialized laboratory, which is often difficult to deploy in rural areas [9–11]. PCR variants, such as multiplex-real time PCR (MRT-PCR), could be useful in rapid diagnostics for TB and TB mutants that are slow and hard to isolate in culture [12].

Recent advances in infectious diseases diagnostic biosensors offer an improvement in precision with a reduction in time and cost [13–16] and the capacity to be used in low resource settings [17]. There are a great variety of biosensor designs and methods of translation. Electrochemical methods for detection are becoming popular in particular in recognition of DNA [18]. With electrochemical reactions used to detect the changes in electrical/conductive properties originated from DNA hybridization, the costs of these detection systems are relatively economical. In the case of DNA biosensors, the probes can include a plethora of substrates allowing the test to become more portable [19, 20].

In this study, we show the design, standardization and evaluation of a fast, cheap and highly sensitive/specific, voltammetric electrochemical biosensor based on electrochemical detection of DNA hybridization for identification of the DNA sequence IS6110 of *Mycobacterium tuberculosis* in both sputum and urine samples.

## Materials and methods

### Design of probes and primers

We use the insert 6110 (IS6110), which is a specific sequence for the detection of M. tuberculosis DNA. IS6110 is a recognized genotypic biomarker for epidemiology and widely distributed in the strains collected worldwide. Due to its unique presence in Mycobacterium complexes and its high level of replication through the entire genome, IS6110 is used as a gold standard biomarker for diagnosis [21, 22]. The first capture probe is designed with the addition of an amine group in 5′ end to immobilize the working electrode, (5′NH3-GTTGTTCTTCCGCAT GAGCTGGACTTTCTGCAATAGGTGGTATGCCTATCCCCTAGAGTCATGTGTAGCTAGGCCA AGTCGCTC-3′). This sequence showed a 100% match with the *M. tuberculosis* complex sequence. The fragment sequence 5′-CAA CAA GAA GGC GTA CTCGAC CTG AAA GAC GTT ATC CAC CAT ACG GAT AGG GGA TCT CAG TAC ACA TCG ATCCGG TTC AGC GAG-3′ was selected as a synthetic positive control sequence for hybridization.

Secondary structure of DNA sequences was evaluated using MFold Web software, with the thermodynamic parameters of 68˚C and 10nM for the saline concentration during the amplification process. The primer sequences to amplify the DNA detection element from sputum were designed with Primer3 software to account for specificity (forward 5′- CAA CAA GAA

GGC GTA CTC GAC CTG A-3´ and reverse 5´-CTC GCT GAA CCG GAT CGA TGT GTA CT-3´). The primers had a melting temperature of 71.0˚C (forward) and 73.4˚C (reverse) with a %GC of 52.0% (forward) and 53.9% (reverse). The capture probe showed a melting temperature of 82˚C.

For detecting M. tuberculosis from urine samples, we used another fragment segment from IS6110 (5' NH3-ACCAGCACCTAACCGGCTGTGGGTAGCAGACCTCACCTATGTGTCGACC TGGGCAGGGTTCGCCTACGTTG3') as capture probe second, previously reported [22]. Also, the synthetic sequence 5´-CAACGTAGGCGAACCCTGCCCAGGTCGACACATAGGTGAGGTCT GCTACCCACAGCCGGTTAGGTGCTGGT-3´ was used as a positive control for the urine samples.

For the PCR amplification of this fragment, we used the primers: Forward:5'-ACCAGCAC CTAACCGGCTGTGG-3´ and Reverse: 5'-GTAGGCGAACCCTGCCCAGGTC-3´. As a negative hybridization control sequence, the non-complementary short synthetic DNA sequence 5 ´-CAA CAA GAA GGC GTA CTC GAC CTG AAA GAC GTT ATC CAC CAT ACG GAT AGG GGA TCT CAG TAC ACA TCG ATC CGG TTC AGC GAG-3´, was used.

## Preparation of the working electrode

The working electrode was assembled in a 2 cm x 0.5 cm glass coated with a thin film of oxide tin-doped fluorine (FTO-glass) (Aldrich, USA). The working electrode was placed in a 1.5 mL tube, with 1 mL of acetone. It was sonicated (Branson 2800) for 10 minutes, followed by a washing step with isopropanol and dried under a flow of nitrogen gas ($N_2$). Subsequently, the electrode was placed in a 1.5 mL tube with 1 mL of isopropanol and sonicated for an additional 10 minutes. Next, the electrode was washed with ethanol and dried with $N_2$. This process of sonication was repeated with 1ml of ethanol, washed with MiliQ water and dried with $N_2$. Finally, it was again sonicated in MiliQ water, washed in MiliQ water and dried with $N_2$.

At the end of the washing process, the FTO-Glass electrode was submerged in a 1.5mL tube with 1 mL of a solution of ammonium hydroxide and hydrogen peroxide in miliQ water (1:1:5) at 60˚C for 10 minutes; followed by 1-hour incubation in a 1.5 mL tube with 500 mL of 2% 3- Aminopropyl triethoxysilane (APTES) (Aldrich, USA) in 95% ethanol at room temperature. Following incubation, the electrode was washed in absolute ethanol, followed by washing with water and drying in $N_2$. Next, the electrode was incubated in a 1mL tube where 0.5ml of a 50 nM Phenylene diisothiocyanate (PDITC) (Aldrich, USA) in N, N dimethylformamide (DMF) (Sigma Aldrich USA) solution was added for 1 hour at room temperature. It was followed by a wash with absolute DMF and finally washed with absolute ethanol, and dried with $N_2$. The DNA capture probe was immobilized by depositing 40 μL of 1 mM DNA in 1 M PBS under an area of approximately 0.5 $cm^2$ of the working electrode, and left to incubate for 1 hour at room temperature, after which it was washed in 1M PBS, and dried with $N_2$. The non-specific sites of the surface were blocked by adding 40 μL of 50 mM ethanolamine in MiliQ water on the FTO-glass electrode conductor surface. After 10 minutes, the electrode was washed with MiliQ water, dried with $N_2$, and stored at 4˚C. With MiliQ water, dried with N2, and stored at 4˚C. The same procedure was performed with the second capture probe but due to its smaller size (69 pb) a final concentration of 1.5 mM was used, in order to compensate and achieve a similar surface density.

## Surface and electrochemical characterization of the working electrode

To achieve quality control during the preparation of the biosensor, morphologic studies of the FTO-glass surface were performed at each step, by scanning electron microscopy at 10,000X magnification in an SEM JEOL Ninja microscope with a discharge voltage of 20 kV. A chemical-elemental analysis of the surface of the electrode was performed by dispersion of X-ray

energy using an OxfordEDX microanalyzer. Raman spectroscopy was performed to obtain a characteristic spectrum of each stage of the biosensor fabrication at 630 nm wavelength using the Raman Spectrometer Xplora (Horiba Scientific). To enhance Raman scattering (SERS), 100 μL of 0.1 mM of colloidal silver nanoparticles was added to the glass surface.

Electrochemical characterization of each stage of the FTO-glass electrode assembly was performed by cyclic voltammetry. We used a potentiostat (Uniscam PG581) with a three-electrode configuration electrochemical cell; reference electrode (Ag/AgCl), work electrode (FTO Glass), and a platinum counter electrode. Voltammetry was conducted in 0.1 M $K_3[Fe(CN)_6]$ / $K_4[Fe(CN)_6]$ redox couple in 1M PBS solution. A voltage window from -400 mV to 1.1 V with a scanning rate of 100 mV/s was used. Additionally, cyclic voltammetry evaluation of the top coat with the DNA capture probe was performed at different scanning rates (10, 20, 50, 100, 150, 200, 250 mV/s). In this case, the working electrode was the FTO glass immobilized with DNA probes. In all cases, repetitions of 5 cycles were performed.

### Extraction of genomic DNA from *M. tuberculosis* strains and sputum of TB patients

Genomic DNA was extracted from *M. tuberculosis* H37Rv obtained from our strain library. For this, *M. tuberculosis* culture was inactivated with nano beads (0.2 mm) in the FastPrep system (FastPrep 24™ 5G). Lysis was performed by adding 500 μL of Buffer Tris EDTA pH8 and heated at 80˚C for 20 minutes. Then 20 μL of proteinase K was added along with 75 μL of 20% SDS, vortexed and placed in a water bath at 65˚C for 3 hours. Vortexing of the sample was done every 30 minutes. 100 μL of CTAB/NaCl was heated to 65˚C and then added to the sample with 100 μL of NaCl. Vortexing was performed until the sample turned white, and then returned to a water bath at 65˚C for 10 minutes. After 10 minutes, 750 μL of phenol:chloroform:Isoamyl alcohol in the ratio 25:24:1 was added to the sample and homogenized. It was centrifuged at 10,000 rpm for 5 min. The supernatant was placed in another 1.5 mL tube, while 750 μL of chloroform: isoamyl alcohol in a 24:1 ratio was added and stirred until the sample was homogenized. The sample was centrifuged again at 10,000 rpm for 5 minutes. The supernatant was removed and washed a second time with chloroform: isoamyl alcohol 24:1. Subsequently, the supernatant was extracted by adding 1 mL of cold absolute alcohol and stirred until the DNA precipitated. A final round of centrifugation at 10,000 rpm for 5 minutes was performed and the supernatant was removed. Then 1 mL of 70% ethanol was added and washing was repeated. Finally, the supernatant was discarded and the precipitate was resuspended in 100 μL TE at pH 8.

To extract the DNA from the sputum of patients with tuberculosis, each sputum sample was vortexed with 500 μL phenol:chloroform:isoamyl alcohol for 30 minutes, followed by 3 cycles of FastPrep (40s, 5.5 speed). Subsequently, 350 μL of binding buffer was added to the crushed sample, then left on ice for 5 min and then centrifuged for 10 min at maximum speed (14 800 RPM). The supernatant was transferred to a high purity filter tube and centrifugated at maximum speed for 1 min. The content of the collection tube was discarded and the filter tube was reinserted in the collection tube. 700 μL of washing buffer was added to the filter tube (directly to the centre) and centrifuged for 30–60 seconds at maximum speed. The content of the collection tube was discarded and subsequently added 50 μL of elution buffer (10 mM Tris Buffer, pH 8.5) and was stored at 4˚ until it was used.

### Evaluation of the biosensor with DNA from *M. tuberculosis* H37Rv reference strain

The extracted DNA was taken to four concentrations: Dilution 1 (160 ng/μL), Dilution 2 (0.016 ng/μL), Dilution 3 (0.00016 ng/μL) and Dilution 4 (0.000016 ng/μL). Dilutions were

done in 1 mM PBS and absorbance at 230 nM was measured. A conventional PCR with 35 cycles was performed at for each of the described DNA dilutions under the following conditions: 95˚C in the denaturing step for 5 minutes, 62˚C in the annealing step for 1 minute and elongation at 72˚C. PCR amplification product was heated to 95˚C for 5 minutes and placed for 1 minute at 0˚C, then mixed with 1 mM PBS in 1:1 volume. 40 μL of this solution was added on the surface of the sensor. The later electrode was cleaned with PBS, dried with $N_2$ and measured in the electrochemical cell under the conditions described above. The concentration of DNA present in the sample was measured by gel electrophoresis.

### Evaluation of the biosensor with sputum samples from TB patients

Sputum samples from patients with tuberculosis and healthy controls were analyzed. Sputum samples were analyzed from 60 patients with tuberculosis confirmed by the microscopic-observation drug-susceptibility (MODS) culture. The 60 TB culture-positive sputum samples were selected according to the bacillary load according to smear (8 smear +++, 9 smear ++, 14 smears + and 16 smears negative). A total of 13 sputum samples from healthy controls confirmed by a negative MODS test were also analyzed in this study.

A standard PCR was performed on each sample under the same conditions mentioned above. 20 μL of PCR product was heated at 95˚C for 5 minutes and placed for 1 minute at 0˚C degrees, then mixed with PBS in 1:1 volume, placing 40 μL of the solution on the surface of the sensor. Later, the electrode was cleaned with PBS, dried with $N_2$ and measured in the electrochemical cell under the conditions described above.

The sampling framework for this study was opportunistic, making use of DNA extracted bank. The DNA was extracted from sputum samples collected in the course of a research study conducted amongst adults with suspected pulmonary tuberculosis in a hospital-based study in Lima. The study had been reviewed and approved by the Institutional Review Board of University Peruana Cayetano Heredia (Lima, Peru).

### Preparation of a urine sample spiked with *M. tuberculosis* genomic DNA and evaluation of the biosensor

Recent studies are demonstrating the presence of *M. tuberculosis* DNA in the urine of TB patients [23, 24]. To evaluate the capacity of our biosensor, to detect *M. tuberculosis* DNA in urine, a urine sample from a healthy control was spiked with *M tuberculosis* DNA. 5 mL of urine with a 10 mM EDTA in Tris-HCl pH 8.5 was spiked with *M. tuberculosis* DNA at different concentrations (100 ng, 1 ng, 10 pg, 100 fg and 1 fg). Each mixture was homogenized by inversion for 5 minutes and stored at 4˚C until use. Subsequently, 5 mL of urine and 10 mL of absolute ethanol were added in a 15 mL tube, leading to incubation at -20˚C for 20 minutes. At the end of the incubation time, the tube was centrifuged at 5000 rpm for 10 minutes after which the pellet was re-suspended with the binding buffer of the high pure DNA extraction kit (ROCHE, Germany). The solution was transferred to a silica column for DNA purification and was finally eluted in 50 μL of elution buffer (10 mM Tris Buffer, pH 8.5).

The DNA that was extracted from each urine sample (2.5 μL) was added to the PCR master mix (0.3 μM forward primer, 0.3 μM backward primer, 1x bright green). The primers used were: Forward: 5′-ACCAGCACCTAACCGGCTGTGG-3´, Reverse: 5′-GTAGGCGAACCCT GCCCAGGTC-3′. In the Thermal cycling consisted of 95˚C for 10 min to activate the Taq DNA polymerase and for 35 cycles of 94˚C in the denaturing step for 2 minutes, 62˚C in the annealing step for 30 seconds and elongation at 68˚C for 60 seconds. Thereafter, the PCR product was heated to 95˚C for 5 minutes and placed for 1 minute at 0˚C degrees, then mixed with 1 mM PBS in 1:1 volume. 40 μL of this solution was added on the surface of the sensor for

1 hour in room temperature to hybridization. After this, the surface of the sensor was washed with PBS, dried with $N_2$ and measured in the electrochemical cell under the conditions described above.

Each sample processed in the electrochemical biosensor was also tested by Real Time-PCR. Briefly, for 10 μl reactions, the final concentrations of each PCR reagent were: 1X bright green PCR Master Mix, 0.3μM forward primer, 0.3 μM reverse primer and 2.5 μl DNA.

The conditions of the Thermal cycling phase consisted of activation at 95˚C for 10 minutes and for 40 cycles of denaturation at 95˚C for 10 minutes and the extension phase at 62˚C for 60 seconds In addition, a Pre-Melting step at 95˚C for 10 seconds and a Melting temperature range 60˚C to 97˚C was added, with a temperature increase of 0.1˚C/s. all reactions were accomplished in the LightCycler Nano thermal cycler (Roche, Germany).

### Data analysis

The response of the sensor to the concentration of the *M. tuberculosis* DNA present in the samples was measured by a response ratio (cathodic current peak of the hybridized sensor / cathodic current peak of the non-hybridized sensor). The response ratio in the biosensor is expected to be inversely related to the concentration of *M. tuberculosis* DNA in the sample. Therefore, lower values of response ratios are expected to be evidence of hybridization with higher concentrations of DNA, and therefore TB positive samples.

We modelled the culture status (TB positive / TB negative) in a simple logistic regression using the electrochemical response ratio as the predictor. The response ratios were compared between the TB culture-negative and the TB culture-positive sputum samples using a T-test.

To estimate a cutoff for the response ratio that can be used to classify a sample as TB positive or TB negative, we compared the distributions of the response ratios between the TB culture-negative and the TB culture-positive Bk negative groups. The sensitivity of the DNA biosensor was estimated for all the TB MODS culture positive, including the TB positive Bk negative group. The specificity was estimated based on the TB culture-negative samples. All statistical tests were performed under a 95% confidence level.

## Results

### Surface and electrochemical characterization of the working electrode

SEM characterization showed a specific superficial morphology of the nude FTO-glass (Fig 1A). After treatment with APTES, the superficial morphology showed to be homogeneously distributed on the FTO-glass (Fig 1B). However, after immobilization of PDITC and DNA layers, about the same magnification showed globular shaped aggregations/clusters (Fig 1C and 1D). X-ray energy dispersion demonstrated the addition of new elements in each of the stages. In the case of the first modification with APTES, the appearance of silicon on the surface was confirmed (Fig 1). With the addition of the PDITC, it was observed the presence of silicon and nitrogen. Elemental analysis of the last layer with DNA showed Nitrogen again.

Raman spectroscopy of the FTO-Glass showed a peak at 610 cm$^{-1}$ as seen from F:O interaction, after which peaks at 456 cm$^{-1}$ appeared, corresponding to the interaction between APTES and the FTO (Fig 2). In the PDITC stage, a steep peak is observed in the 1300 cm$^{-1}$ of the Raman spectrum (Fig 2). In the DNA stage, a steep peak is observed in the 1085 cm$^{-1}$ of the Raman spectrum (Fig 2).

The cyclic voltammetry curves of the functionalization stages showed changes in current peaks and potential differences (Fig 3A). In the first stage (FTO-glass-nude) the two redox peaks showed a difference of 70mV indicating adequate electronic transfer. Post-modification with the first APTES monolayer exhibits an anodic peak to 0.13 V. The PDITC stage showed

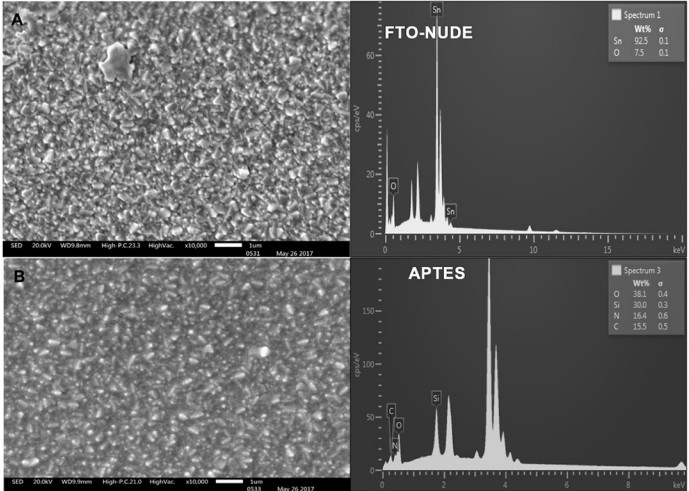
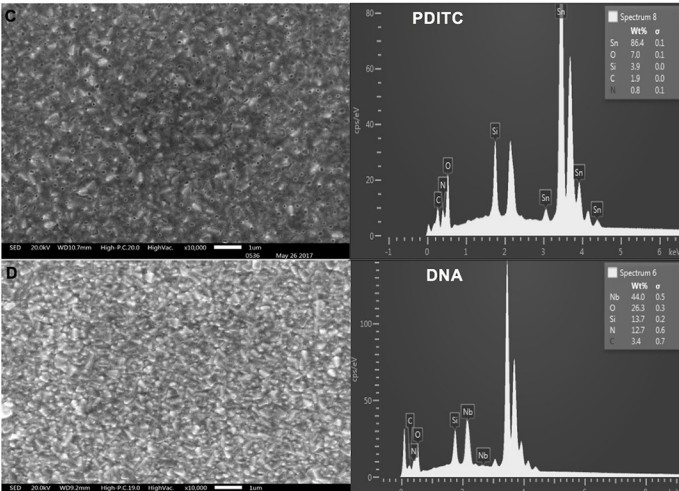

**Fig 1.** (A) SEM Microscopy SEM 10000x 15 kV FTO-GLAS Clean, FTO-GLASS EDX Analysis (B) Microscopy SEM 10000x 15 kV FTO-GLAS-APTES with their respective diffraction in FTO- GLAS-APTES; (C). Microscopy SEM 10000x 15 kV FTO-GLASS-APTES-PDITC and Diffraction; (D). Microscopy SEM 10000x 15 kV FTO-GLAS-APTES-PDITC-DNA and diffraction.

an anode peak at $1.58 \times 10^{-4}$ μA, a cathodic peak at $-1.39 \times 10^{-4}$ μA, and difference of potentials at 0.16 mV similar to the APTES stage. When the capture DNA probe was immobilized and blocked in ethanolamide, it generated current peaks at anode $1.19 \times 10^{-4}$ μA and cathode $-1.15 \times 10^{-4}$ μA, with a potential difference of 0.22 V. With the hybridization of 50 nM of target DNA, the current had an anodic peak $7.61 \times 10^{-5}$ μA and a cathodic peak $-9.75 \times 10^{-5}$ μA. The sweep rate change generates a change in the size of the redox peaks indicating that our biosensor presents a reversible behavior, meaning that the change in the current peak is intimately related to the diffusional movement of the ions (Fig 3B).

## Evaluation of the biosensor with synthetic positive and negative hybridization control sequences

A cathode current peak was observed in response to the synthetic positive control sequence, giving a difference of 55 μA to the sensor response with the non-complementary sequence and 56.7 μA to the negative control respectively (Fig 4). The response of the sensor current to different DNA concentrations showed a logarithmic trend (Fig 5), as shown by the equation that best fits the data: Intensity cathodic peak (A) = $-6 \times 10^{-6} \ln$ (DNA-concentration [nM]) $+0.0001$ with $R^2 = 0.993$ in the range of 1 nM to 500 nM. The curve made with the DNA sequence for urine samples offers linearity with R2 = 0.956 in a 1 nM to 100 nM range (Fig 6 present the average value of 4 identical repetitions).

## Evaluation of the biosensor with DNA from *M. tuberculosis* H37Rv reference strain

Subsequent dilutions of the PCR amplification product from the DNA extracted from *M. tuberculosis* strains showed concentrations ranging from 0.016 pg/μL to 160 ng/μL. The SDS PAGE electrophoretic analysis of the PCR product appeared positive in two of the four dilutions ($2 \times 10^{-2}$ and $2 \times 10^{-5}$) but appeared negative in the dilution $2 \times 10^{-7}$ and $2 \times 10^{-8}$ (Fig 7). However, the biosensor response ratio for these dilutions appeared to be less than 1, showing evidence of positivity for TB (Fig 7).

## Evaluation of the biosensor with sputum samples from TB patients

The response ratios of all the samples processed confirm a significant difference between the values corresponding to the TB culture-positive sputum samples (N = 42, mean = 0.75, SD = 0.20) and the TB culture-negative samples (N = 13, mean = 1.28, SD = 0.31) (P<0.0001) (S1 Table, Fig 8). When considering only the TB culture-positive/smear-negative (N = 16), the response ratio of this group (N = 16, mean = 0.83, SD = 0.18) was also significantly lower than TB negative (N = 13) sputum samples (P<0.0001). The response ratio was able to explain 68.21% of the variability of the Log(Odds) of TB culture positivity (OR = $1.26 \times 10^{-8}$, P = 0.02). The logistic regression to model the TB culture status (positive/negative) showed a 0.9698 area under the ROC curve, with an overall sensitivity of 97.62% and a specificity of 92.31% corresponding to a probability cutoff of 0.74 (response ratio cutoff = 1.0) (Fig 9). When only considering the TB culture-positive/acid-fast smear-negative sputum samples (N = 16), fifteen of them were classified positive by the biosensor (sensitivity for acid-fast smear-negative, TB culture-positive = 93.8% (15/16)). One of the thirteen TB negative samples was erroneously classified as positive by the biosensor (specificity = 92.3% (12/13)), (Fig 8, S1 Table).

## Evaluation of the biosensor in urine sample spiked with *M. tuberculosis* genomic DNA

The evaluation of the urine sample spiked with 100 ng, 1 ng, 10 pg, 100 fg and 1 fg of MTB DNA showed a significant difference between positive and negative controls, with RT-PCR confirming the presence of DNA in each of the samples (Fig 8).

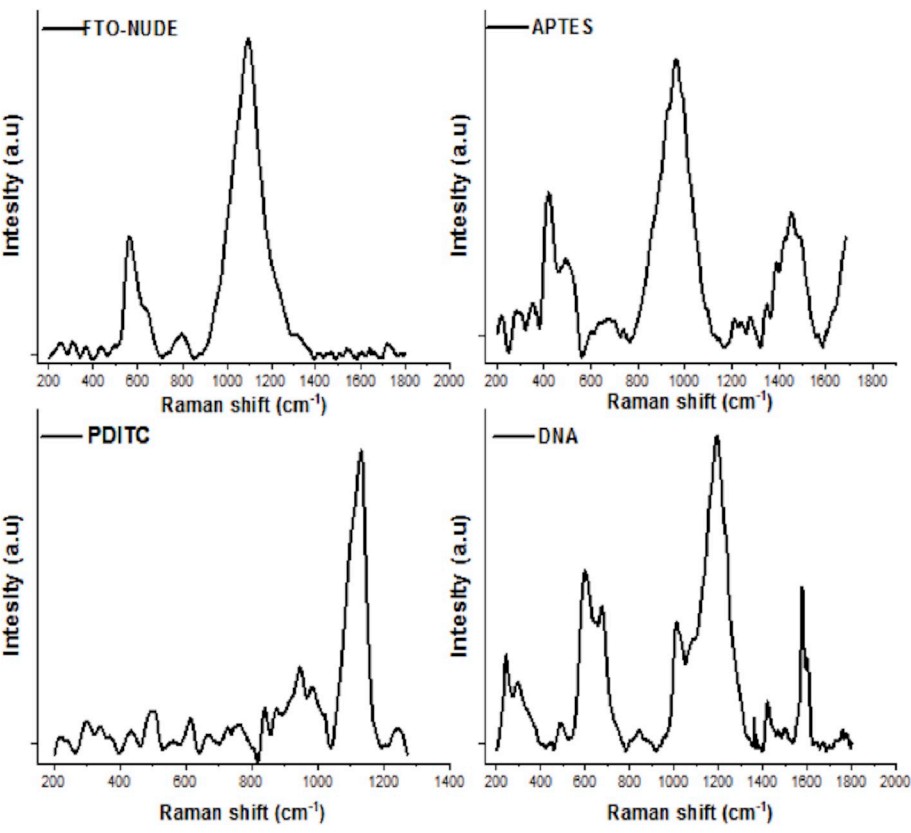

**Fig 2. Raman shift of interaction with each monolayer; FTO-nude, this the first step for modification of the sensor, followed by monolayers of APTES, PDITC and DNA, showing an interaction between them demonstrating adequate coupling.**

Our biosensor allowed to recognize specific MTB DNA sequences without being affected by the background solution coming from the urine, demonstrating high sensitivity and reproducibility of the sensor.

## Discussion

The biosensor developed in this study was able to detect DNA sequences of *M. tuberculosis* in very low concentrations, in both sputum and urine with different extraction methods without this representing a limitation in the differentiation of negative and positive samples.

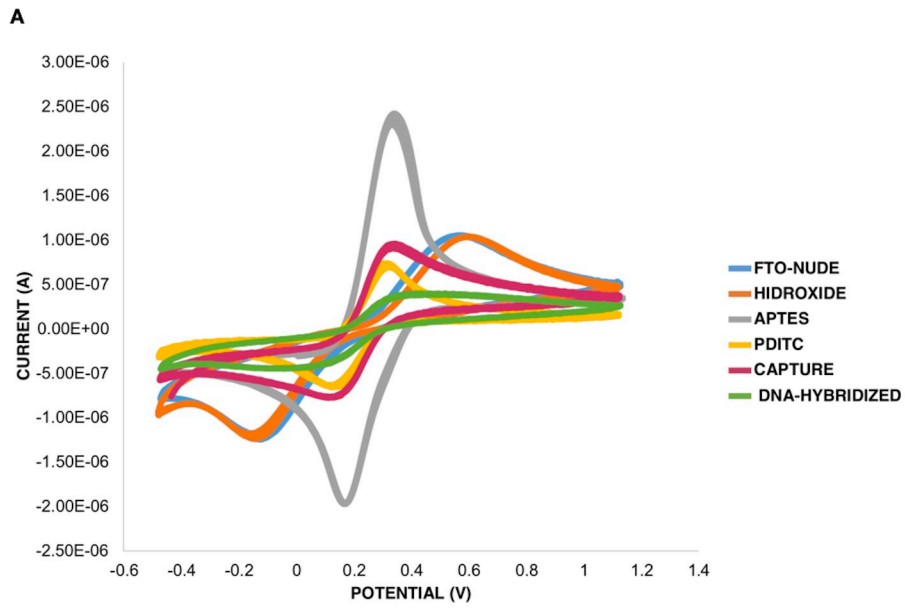

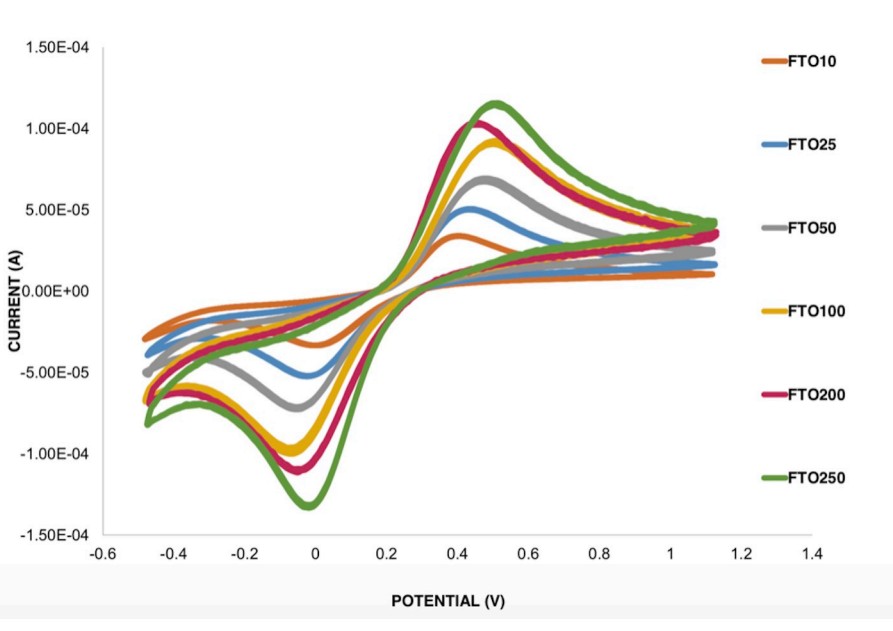

**Fig 3.** (A) Voltagrams of each stage of immobilization in an electrochemical cell with a three-electrode configuration; Reference electrode (Ag / AgCl 1.0 M), Counter-electrode (Platinum), Working electrode (FTO-MODIFIED) Ferri / Ferro 0.1 M in Buffer PBS 0.1 M, in a potential window -0.400 to 1.100 V, at a rate of 100 mV/s. (B) Voltagrams of the different scanning rates using the FTO-DNA capture electrode, ranging from 10 mV/s to 250 mV/s.

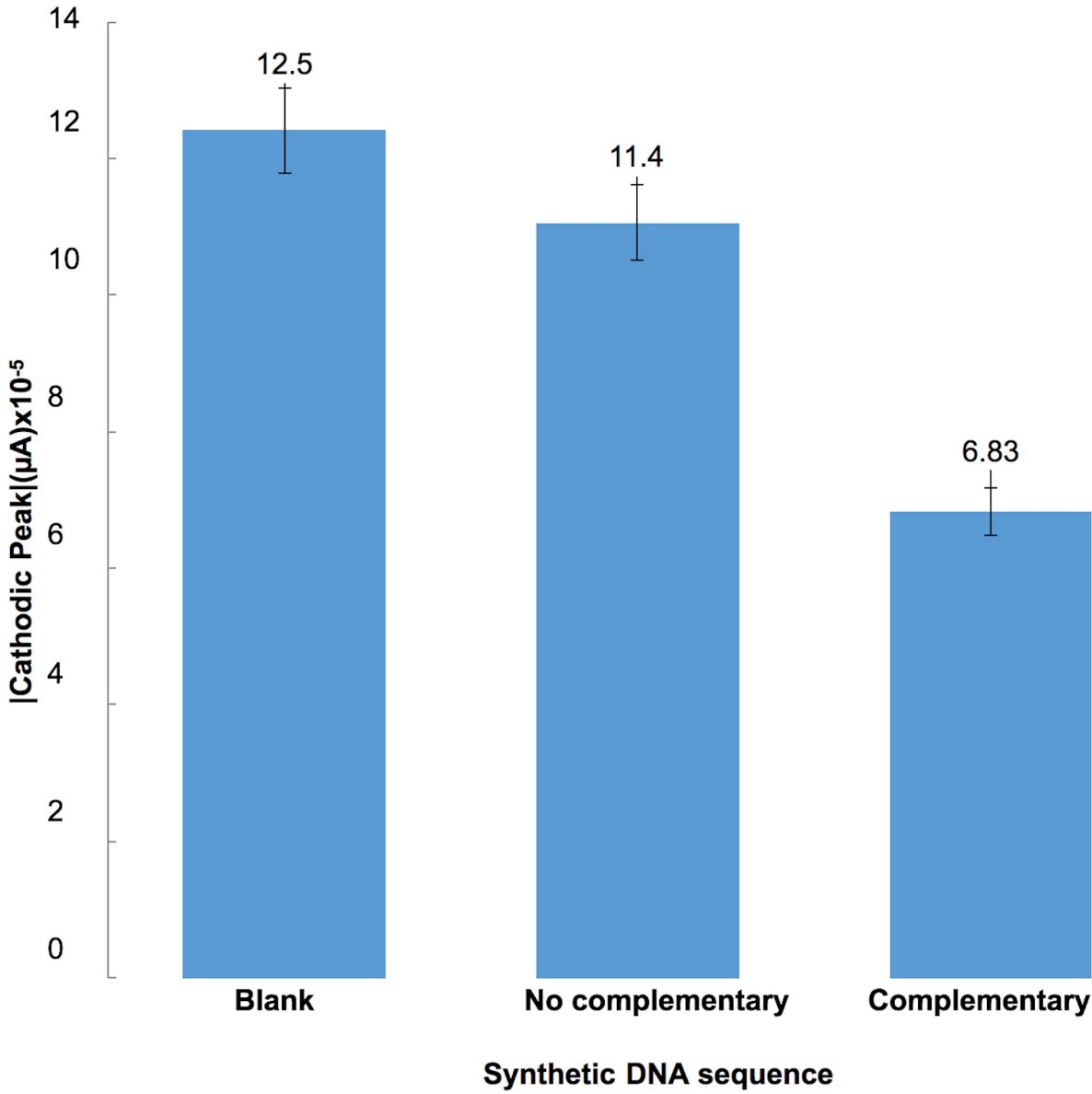

**Fig 4. Sensor response represented by the behavior of the absolute value of the cathodic peaks towards samples without synthetic DNA (BLANK), samples with 50nM of non-complementary synthetic DNA sequence and samples with 50nM of complementary synthetic DNA sequence.**

The sputum samples from TB patients showed a cut-off point of 0.74 response ratio. Furthermore, the sensor reached 97.62% sensitivity and 92.3% specificity, suitable for use it as a diagnostic tool. In the other hand, our method allows the recognition of amplicons from a conventional PCR. In contrast to SDS-PAGE or real time PCR, our technique is free of labels/tags and dyes that increase the cost of the test. It is important to highlight that the TB culture-positive sputum samples used (N = 42), were Bk negative and were contrasted against TB culture negative samples (N = 13). This is the most adverse scenario for testing the biosensor, because of the low bacilli load and consequently the low DNA concentration available. Moreover, the biosensor using synthetic DNA control tested with urine showed a linear trend in range of

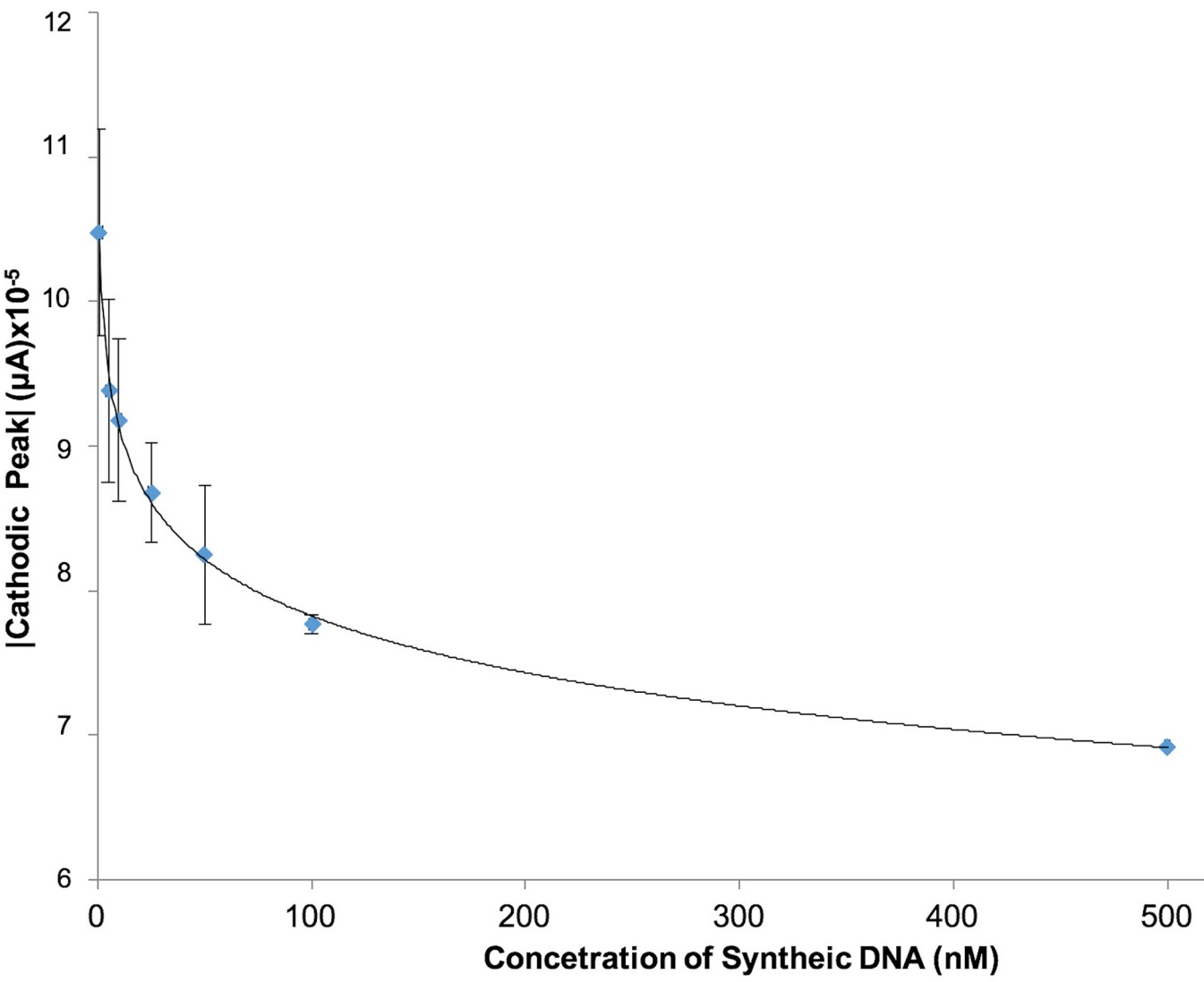

**Fig 5. Sensor calibration curve using the absolute value of cathodic current peak versus target DNA concentration in a range 1 nM to 500 nM, showing a logarithmic trend.**

1nM to 100nM, therefore, our biosensor presents a linear dynamic range from 1nM to 100nM in the synthetic sample.

The response ratio of our biosensor using synthetic DNA control sequences showed a logarithmic trend with the DNA concentration in the range 1 nM to 500 nM, where the biosensor was able to differentiate between TB positive and TB negative samples with a detection limit of 1.5 ng/$\mu$L of synthetic DNA.

In addition, for genomic DNA with a previous PCR amplification, the biosensor showed a high sensitivity of detection of 16 fM (16 x $10^{-15}$ M). The limits showed are close to previously reported values [25, 26]. In contrast, our study used cheap and non-complex materials. The changes in the voltammogram in each of the layers added on the FTO-GLASS are theoretically related to the constant of the electron transfer rate and the electron transfer resistance [27]. In the case of the APTES, the reduction of the potential difference is attributed to a higher concentration of the anionic probes $[Fe(CN)_6]^{3-/4-}$ due to their strong affinity with the polycarbonate layer, since the amino groups of the APTES are protonated ($NH_3^+$) in aqueous solution.

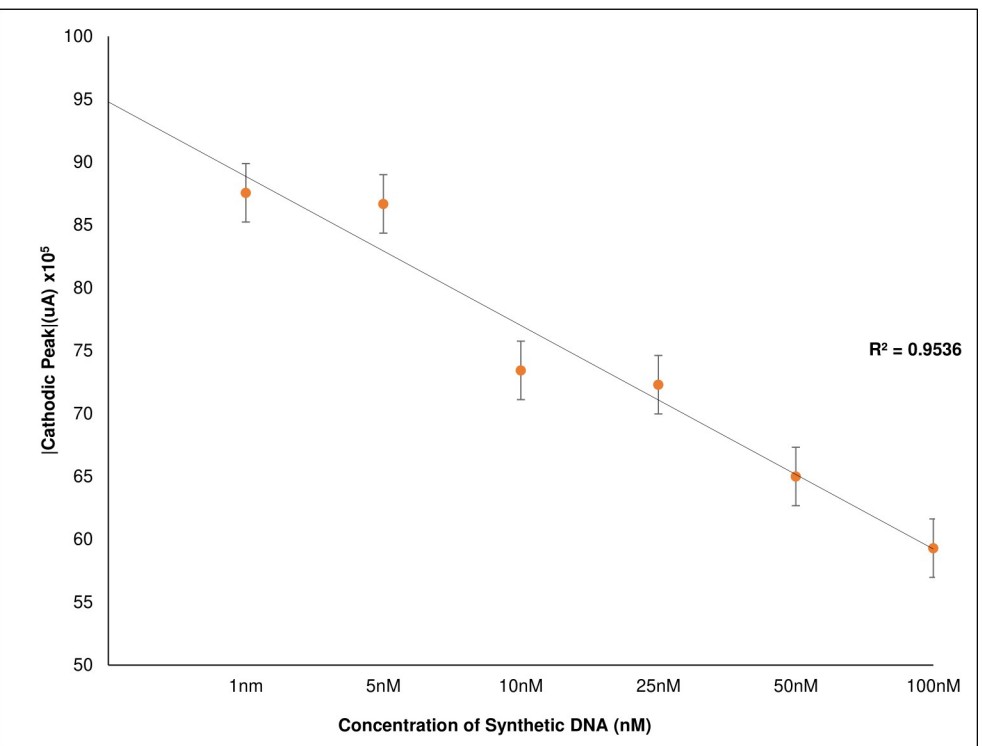

**Fig 6. Sensor calibration curve using the absolute value of cathodic current peak versus target DNA concentration in range 1nM to 100nM, showing a lineal trend.**

The PDITC aggregation showed a peak-reduction due to the PDITC covering, producing a more stable system and limiting an efficient ion-transfer over the surface.

The rate scan variation in the immobilized DNA step has shown a diffuse behavior, indicating that mass transfer is reflected by the variation in $K_3[Fe(CN)_6]/K_4[Fe(CN)_6]$ redox couple concentration. When the DNA probe is immobilized on the surface of the electrode, the redox peak current decreases, changing into a quasi-reversible system. After hybridization with the target DNA, the current peak of the modified electrode was further reduced, indicating that double-stranded (hybridized) DNA formed an electron transfer and a mass transfer blocking layer [28]. The most widely used method to assess the extent of electrode surface coverage by DNA is based on cyclic voltammetry of ferricyanide [28].

Previous studies found that the ideal size of the sequence probe for immobilization in a biosensor is 50 nucleotides, in order to prevent non-specific hybridizations and steric interference [29]. Although the efficiency could be affected by the size heterogeneity of the DNA sequences present in the samples [30, 31]. Our study showed that a probe sequence of 84 nucleotides performed better in detecting *M. tuberculosis* DNA in real sputum samples, however, the best response for the detection of *M. tuberculosis* DNA in urine corresponded to the 69 nucleotides probe. It is likely that this probe did not form significant secondary structures and that its size generated an increase in the signal without compromising sensor specificity.

The described biosensor recognizes the hybridization of DNA by changes of its electrochemical properties. This causes an inhibition of the reaction on the surface of the electrode associated with variations in the surface structure and electrical conductivity [32]. With measurements made by SEM microscopy and Raman spectroscopy, this variation is associated with the layers where an FTO morphological surface changes are observed in every process

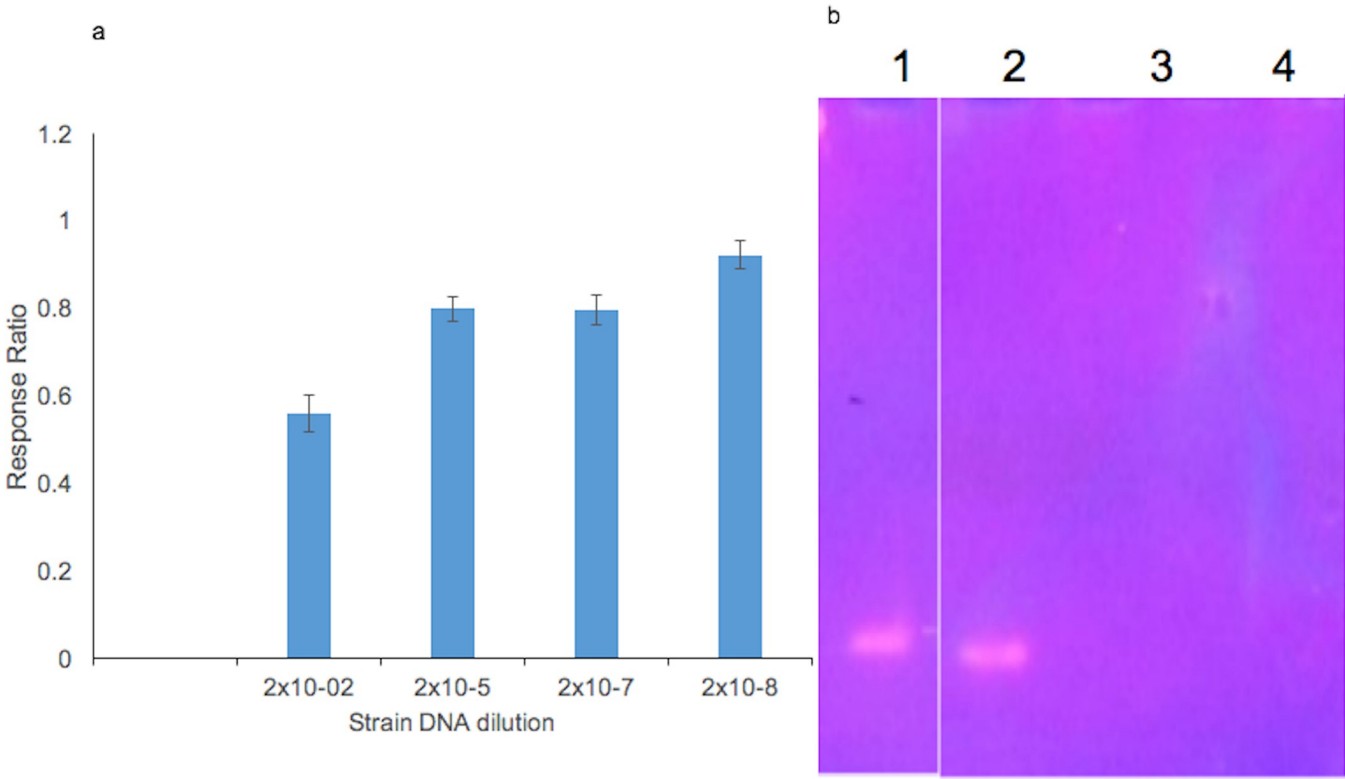

**Fig 7.** (A) Ratio of the sample signal (DNA) divided by the baseline signal (EA) of the concentrations, 160 ng/mL ($2.00 \times 10^{-2}$), 0.016 ng/mL ($2.00 \times 10^{-6}$), 0.00016 ng/mL ($2.00 \times 10^{-8}$) and 0. 0.000016 ng/mL ($2.00 \times 10^{-9}$). (B) 12% acrylamide gel of the PCR amplicons at dilutions 1 ($2.00 \times 10^{-2}$), 2 ($2.00 \times 10^{-6}$), 3 ($2.00 \times 10^{-8}$) and 4 ($2.00 \times 10^{-9}$).

step as shown before [33]. This is supported by the X-ray scattering analysis. In the case of Raman spectroscopy, the aggregation of the peaks was in accordance with layer aggregation and bond stability [34, 35].

In conclusion, this study presents two biosensors that are effective for the detection of 16 fM of *M. tuberculosis* DNA in sputum sample and 1 fM of *M. tuberculosis* DNA spiked in a

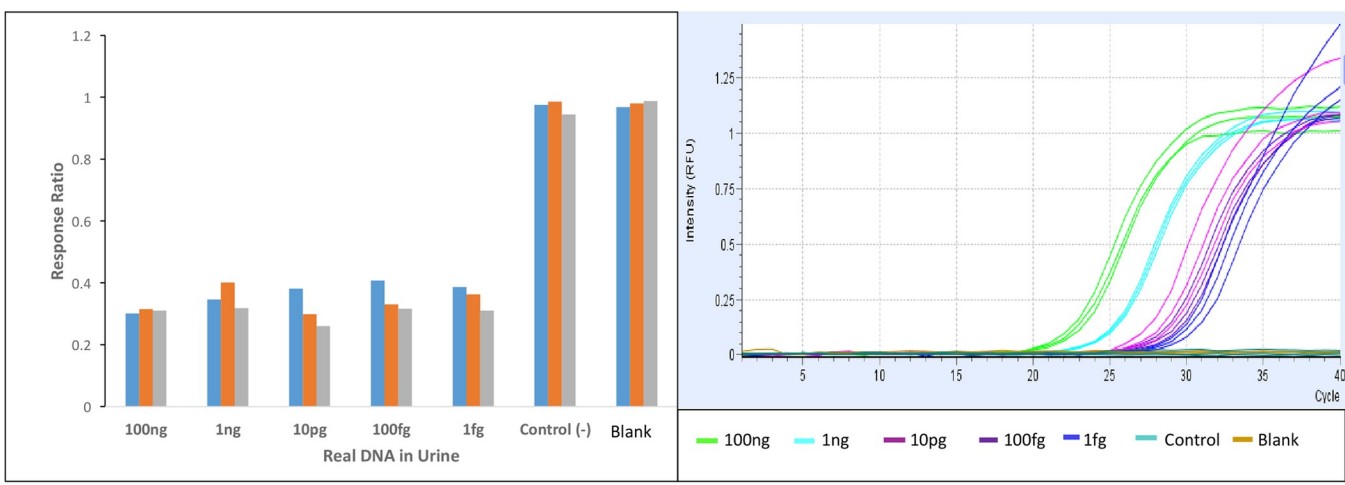

**Fig 8.** (A) Three "Spikes" urines were evaluated containing 100 ng, 1 ng, 10 pg, 100 fg and 1 fg of MTB DNA. It was compared with real time PCR.

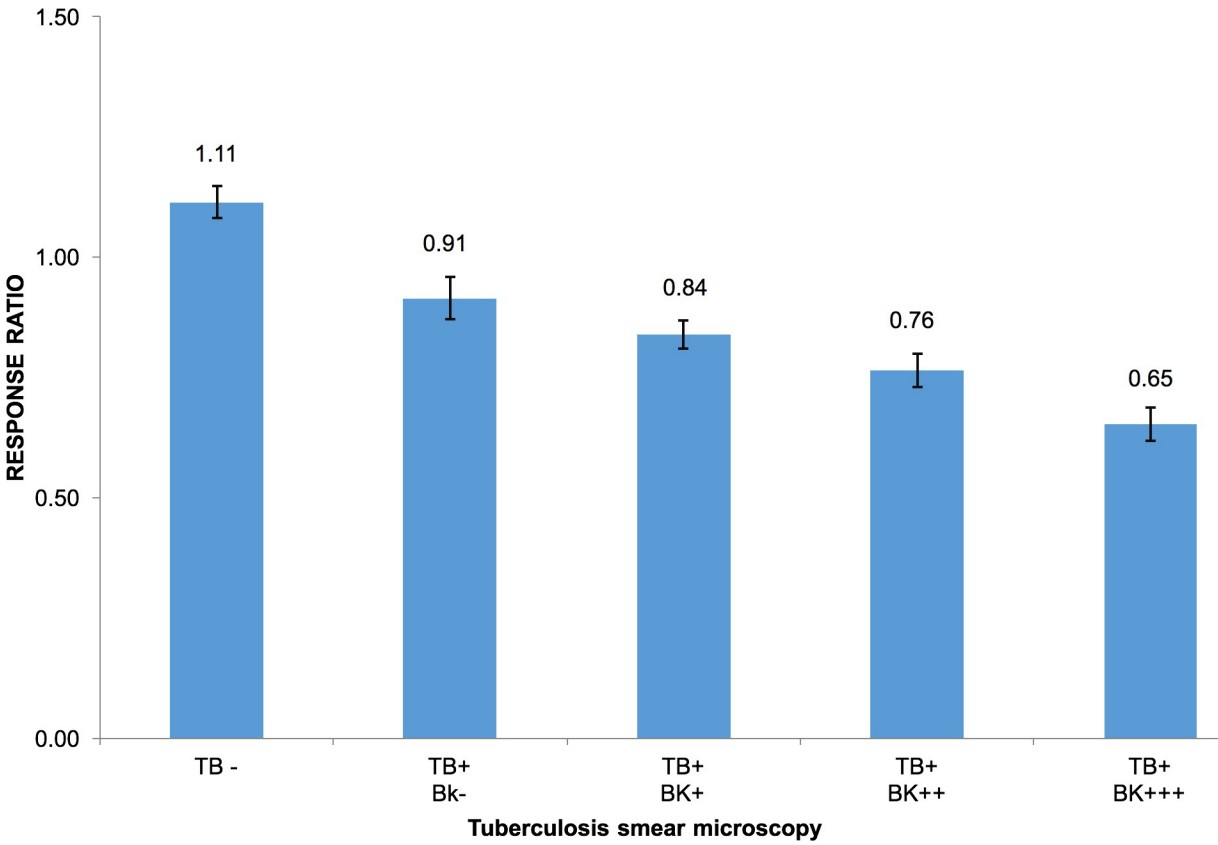

**Fig 9. The response ratio of the sputum samples according to the bacillary load presented in the sputum smear (BK).**

urine sample with a high sensitivity largely exceeding the sensitivity of conventional PCR and SDS PAGE. All these advantages together with the low cost and the relatively brief process completion time, make this a promising assay for TB screening and a potential new tool in the fight against tuberculosis.

## Supporting information

**S1 Table.**
(DOCX)

**S1 Raw images.**
(TIFF)

## Acknowledgments

We acknowledge Nehal Naik for his help in the edition of the first version of this manuscript.

## Author Contributions

**Conceptualization:** Patricia Sheen, Mirko Zimic.

**Formal analysis:** Daniel Rueda, Mirko Zimic.

**Funding acquisition:** Patricia Sheen, Mirko Zimic.

**Investigation:** Daniel Ramos-Sono, Raúl Laureano, Patricia Sheen, Mirko Zimic.

**Methodology:** Daniel Ramos-Sono, Raúl Laureano, Adolfo La Rosa, Jesús Ruiz, Raúl León, Patricia Sheen, Mirko Zimic.

**Project administration:** Patricia Sheen.

**Resources:** Adolfo La Rosa, Jesús Ruiz, Raúl León, Patricia Sheen.

**Supervision:** Patricia Sheen, Mirko Zimic.

**Validation:** Patricia Sheen.

**Writing – original draft:** Daniel Ramos-Sono.

**Writing – review & editing:** Robert H. Gilman, Adolfo La Rosa, Jesús Ruiz, Raúl León, Patricia Sheen, Mirko Zimic.

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
