## [Decision Letter · Decision Letter 0]

12 Jun 2020

PONE-D-20-08686

An electrochemical biosensor for the detection of Mycobacterium tuberculosis DNA from sputum and urine samples.

PLOS ONE

Dear Dr. Zimic,

Thank you for submitting your manuscript to PLOS ONE. After careful consideration, we feel that it has merit but does not fully meet PLOS ONE’s publication criteria as it currently stands. Therefore, we invite you to submit a revised version of the manuscript that addresses the points raised during the review process.

Please fully address the comments by the two reviewers below.  In particular please be sure that the biosensor figures of merit (e.g., sensitivity, response time, selectivity, reproducibility) are clearly stated in your manuscript.  For example, one of the reviewers had concerns regarding the dynamic sensing range of the sensor.

We look forward to receiving your revised manuscript.

Kind regards,

Jonathan Claussen

Academic Editor

PLOS ONE

Journal Requirements:

Additional Editor Comments (if provided):

Reviewers' comments:

Reviewer's Responses to Questions

**Comments to the Author**

1. Is the manuscript technically sound, and do the data support the conclusions?

Reviewer #1: Yes

Reviewer #2: Partly

2. Has the statistical analysis been performed appropriately and rigorously? 

Reviewer #1: Yes

Reviewer #2: No

3. Have the authors made all data underlying the findings in their manuscript fully available?

Reviewer #1: Yes

Reviewer #2: No

4. Is the manuscript presented in an intelligible fashion and written in standard English?

Reviewer #1: Yes

Reviewer #2: Yes

5. Review Comments to the Author

Reviewer #1: -Overall, authors demonstrate a novel technique using E-DNA sensors for improved detection of TB DNA in urine and sputum. Results show good specificity and sensitivity. However, this sensor seems slightly qualified in that it uses PCR amplified DNA from patient samples limiting benefits of point-of-care diagnostics to rural communities lacking easy access to PCR and other laboratory instrumentation.

-Concise, well informed introduction.

-Very thorough cleaning/examining process of electrodes.

Areas for minor revision:

-First Paragraph (methods): line 2---Authors state they amplified IS6110 using generic biomarkers. This is vague and explanation could be improved.

-Second paragraph (methods): line 2---Specifying the temp/saline concentrations would be productive to let other labs confirm/repeat experiment.

-Why did the authors use two different concentrations between the first and the second probe (1mM DNA vs 1.5mM DNA). Please explain inconsitency.

-Was the elution buffer used for their DNA purification from urine samples specified?

-The determined cutoff value for response ratio seems limited in trials as it was only based on TB positive Bk negative vs TB-negative cultures from this singular sample pool. This could be added to discussion.

-Figure 6: how many trials is this representing? Is it the average of all of the trials?all of the trials? More info needed.

Grammatical/other minor errors:

-2nd paragraph (intro), 5th line--duration spelled with two "n"s.

-Paragraph 4 (intro): line 2---"electrochemical biosensor based electrochemical detection…" Add "based on"

-Paragraph 4 (intro): line 3---Might be wrong, but after looking it up and looking at other areas of the --paper it seems the authors meant: IS6110 vs IS6610.

-inconsistencies in formatting in-text citations. (see paragraph 4 of methods, line 3---used square brackets instead of parenthesis.)

-When using their phenol cholorform isoamyl alcohol solution in a sentence they might want to be consistent and keep it phenol:cholorform:isoamyl:alcohol solution.

-Bold-face sub-heading "Evaluation of the biosensor with sputum samples from TB patients".

-Citations 16-18 are missing volume numbers and have dates placed twice instead.

-Update citation 24 to be consistent in formatting (no title/journal volume)

Reviewer #2: The authors report a sensitive electrochemical method for detecting the DNA of M. tuberculosis in sputum and urine samples. In the field of tuberculosis detection, highly sensitive detection is very important, and the author has done a lot of work in biology. However, from the perspective of developing biosensors, reviewers believe that there are many aspects of the manuscript that need to be carefully revised. Therefore, this work is not recommended to publish without a major revision in at least the following aspects.

1. First, the data should support the conclusion. For example, how does the author prove the stability and reliability of the method? From the data (Figure 6), there is a lack of experiments and analysis of these two points. The authors should notice that the R-squared value represents the quality of the fit, not the quality of the method. Please mark all data points and errors in Figure 6.

2. The author should clearly explain their method by adding some comparison and analysis of this method with other recognized standard methods.

3. The dynamic range of the test is unclear.

In addition, there are some grammatical and spelling errors. The author should carefully correct these errors.

6. PLOS authors have the option to publish the peer review history of their article (what does this mean?). If published, this will include your full peer review and any attached files.

Reviewer #1: No

Reviewer #2: No

---

## [Author Response · Author response to Decision Letter 0]

12 Aug 2020

author 1:Thank you for your valuable comments, we have corrected each one of them, taking special care in the spelling.

author 2: Thank you for your valuable comments, we took your advice to improve our article, we had special care in the part of reproducibility, sensitivity and dynamic range.

---

## [Decision Letter · Decision Letter 1]

8 Oct 2020

An electrochemical biosensor for the detection of Mycobacterium tuberculosis DNA from sputum and urine samples.

PONE-D-20-08686R1

Dear Dr. Zimic,

We’re pleased to inform you that your manuscript has been judged scientifically suitable for publication and will be formally accepted for publication once it meets all outstanding technical requirements.

Kind regards,

Olivier Neyrolles

Section Editor

PLOS ONE

Reviewers' comments:

Reviewer's Responses to Questions

**Comments to the Author**

1. If the authors have adequately addressed your comments raised in a previous round of review and you feel that this manuscript is now acceptable for publication, you may indicate that here to bypass the “Comments to the Author” section, enter your conflict of interest statement in the “Confidential to Editor” section, and submit your "Accept" recommendation.

Reviewer #1: All comments have been addressed

2. Is the manuscript technically sound, and do the data support the conclusions?

Reviewer #1: (No Response)

3. Has the statistical analysis been performed appropriately and rigorously? 

Reviewer #1: (No Response)

4. Have the authors made all data underlying the findings in their manuscript fully available?

Reviewer #1: (No Response)

5. Is the manuscript presented in an intelligible fashion and written in standard English?

Reviewer #1: (No Response)

6. Review Comments to the Author

Reviewer #1: (No Response)

7. PLOS authors have the option to publish the peer review history of their article (what does this mean?). If published, this will include your full peer review and any attached files.

Reviewer #1: No

---

## [Editor Report · Acceptance letter]

12 Oct 2020

PONE-D-20-08686R1 

An electrochemical biosensor for the detection of *Mycobacterium tuberculosis* DNA from sputum and urine samples. 

Dear Dr. Zimic:

I'm pleased to inform you that your manuscript has been deemed suitable for publication in PLOS ONE. Congratulations! Your manuscript is now with our production department. 

Kind regards, 

on behalf of

Dr. Olivier Neyrolles 

Section Editor

PLOS ONE